# EGFR Pathway Expression Persists in Recurrent Glioblastoma Independent of Amplification Status

**DOI:** 10.3390/cancers15030670

**Published:** 2023-01-21

**Authors:** Andrew Dhawan, Venkata S. K. Manem, Gabrielle Yeaney, Justin D. Lathia, Manmeet S. Ahluwalia

**Affiliations:** 1Neurological Institute, Cleveland Clinic, Cleveland, OH 44195, USA; 2Faculty of Pharmacy, Laval University, Quebec City, QC G1V 0A6, Canada; 3Department of Mathematics & Computer Science, University of Quebec at Trois-Rivières, Trois-Rivières, QC G8Z 4M3, Canada; 4Quebec Heart & Lung Institute Research Center, Quebec City, QC G1V 4G5, Canada; 5Department of Anatomic Pathology, Cleveland Clinic, Cleveland, OH 44195, USA; 6Lerner Research Institute, Cleveland Clinic, Cleveland, OH 44195, USA; 7Rose Ella Burkhardt Brain Tumor and Neuro-Oncology Centre, Cleveland Clinic, Cleveland, OH 44195, USA; 8Miami Cancer Institute, Baptist Health South Florida, Miami, FL 33143, USA

**Keywords:** glioblastoma, EGFR, recurrent glioblastoma, RNA-seq, temozolomide resistance

## Abstract

**Simple Summary:**

We compared gene expression in matched primary and recurrent glioblastoma samples, with a focus on those harbouring extra copies of the gene called EGFR (amplified samples). Our results show that in the setting of newly diagnosed glioblastoma, EGFR-amplified compared to EGFR non-amplified tumours display altered gene expression in the EGFR pathway, but this distinction is lost in the setting of recurrent disease. We validated this finding in an independent dataset. EGFR pathway overexpression may be a common mechanism underlying glioblastoma recurrence.

**Abstract:**

Background: Glioblastoma mortality is driven by tumour progression or recurrence despite administering a therapeutic arsenal consisting of surgical resection, radiation, and alkylating chemotherapy. The genetic changes underlying tumour progression and chemotherapy resistance are poorly understood. Methods: In this study, we sought to define the relationship between EGFR amplification status, EGFR mRNA expression, and EGFR pathway activity. We compared RNA-sequencing data from matched primary and recurrent tumour samples (*n* = 40 patients, 20 with EGFR amplification). Results: In the setting of glioblastoma recurrence, the EGFR pathway was overexpressed regardless of EGFR-amplification status, suggesting a common genomic endpoint in recurrent glioblastoma, although EGFR amplification did associate with higher EGFR mRNA expression. Three of forty patients in the study cohort had EGFR-amplified tumours and received targeted EGFR therapy. Their molecular subtypes and clinical outcomes did not significantly differ from patients who received conventional chemotherapy. Conclusion: Our findings suggest that while the EGFR amplification may confer a unique molecular profile in primary glioblastoma, pathway analysis reveals upregulation of the EGFR pathway in recurrence, regardless of amplification status. As such, the EGFR pathway may be a key mediator of glioblastoma progression.

## 1. Introduction

In the setting of recurrent or progressive glioblastoma, re-resection and re-biopsy are generally not preferred options due to the diffuse nature of the tumour, patient comorbidities, and potentially poor functional status. As such, sufficient numbers of clinical samples are not available to answer questions about the genomic behaviour of glioblastoma during the course of therapy. While an understanding of molecular landscapes of primary glioblastoma has been established by a number of large cohorts, such as The Cancer Genome Atlas (TCGA), Pan-Cancer Analysis of Whole Genomes (PCAWG), and Chinese Glioma Genome Atlas (CGGA), there has been a paucity of studies examining the manner in which this landscape is altered after treatment and in the setting of therapeutic resistance [1,2]. With regards to changes in mutational burden and chromosomal integrity, the Glioma Longitudinal AnalySiS (GLASS) consortium has begun the characterisation of the selective pressures of treatment on the glioma genome [1]. That is, the GLASS study has shown that many driver genes in glioma are retained in recurrence, and among the known glioma molecular subtypes, there are differing rates of aneuploidy and alkylating-agent-induced hypermutation at recurrence [1]. More recently, it has been shown that hypermutation may paradoxically confer a unique degree of immunosuppression, perhaps precluding the use of immunotherapies such as checkpoint inhibition, imparting clinical relevance to these findings [2].

Among primary glioblastoma, amplification of the epidermal growth factor receptor (EGFR) gene has emerged as both a key driving event and prognostic factor, occurring in roughly 57% of all primary adult glioblastomas. EGFR-amplified tumours have been shown to be more aggressive and may carry a vulnerability towards targeted EGFR therapies [3,4]. As a result, clinical trials in EGFR-amplified patients of anti-EGFR therapies have been performed (INTELLANCE1—NCT02573324, INTELLANCE2—NCT02343406) but have not yet shown sustained clinical benefit in primary glioblastoma [5]. Gefitinib, in a phase II trial, showed uptake and efficacy in dephosphorylating EGFR in human glioblastoma samples, but no change in downstream EGFR pathway activity. That is, EGFR pathway activity may be independent of its genetic amplification [6].

Here, we characterised 80 glioblastoma samples comprising 40 matched pairs of primary and recurrent tumours. This represents among the largest cohorts to date with RNA sequencing of matched primary-recurrent tumours and robust clinical annotation. We examined mRNA and gene pathway expression between primary and recurrent tumours. We showed key differences in gene expression between primary and recurrent tumours and demonstrated that EGFR-amplified and non-amplified tumours both displayed increased EGFR pathway activity in the setting of recurrence.

## 2. Materials and Methods

### 2.1. Sample Collection and Clinical Information

Glioblastoma samples were collected from surgical specimens from the Cleveland Clinic, in patients with a new diagnosis of primary brain tumour who underwent resection and had re-operation, either for re-resection or biopsy at the time of recurrence. Tumours were diagnosed as glioblastoma using pre-WHO 2016 criteria. Patients were screened for a Phase 1 clinical trial (NCT01800695) for the EGFR antibody–drug conjugate ABT-414. Not all patients received this drug. All data were de-identified for the purposes of this study. Clinical information was collected for pre-selected variables by way of retrospective chart review. All study data collection was approved by the Institutional Review Board of the Cleveland Clinic Foundation.

### 2.2. FISH Evaluation of EGFR Amplification

Fluorescence in situ hybridisation (FISH) was performed by a central laboratory for the 80 glioblastoma samples in the same manner as Lassman et al., as samples presented here are a further characterised subset of these [7]. Full methods can be found in Lassman et al. Briefly, a tumour was considered EGFR-amplified when there was focal EGFR gene amplification defined as a EGFR/CEP7 (chromosome enumeration probe to centromere of chromosome 7) ratio greater than or equal to 2 in at least 15% of recorded cells. Tumours with polysomy for chromosome 7 (excess copies of the entire chromosome defined as CEP7/EGFR < 2 and CEP7 copy number > 3) but without focal amplification of the EGFR gene ≥ 15% were considered to be EGFR non-amplified.

### 2.3. Nucleic Acid Extraction, RNA Sequencing

Nucleic acid extraction followed by RNA sequencing was performed on each of the 80 glioblastoma samples to study their transcriptomic characteristics. Samples presented here were originally described and sequenced in the manuscript by Lassman et al. [7]. Library preparation was performed with 1–50 ng of total RNA. cDNA was prepared using the SeqPlex RNA Amplification Kit (Sigma-Aldrich, St. Louis, MO, USA) per manufacturer’s protocol, and Illumina sequencing adapters were ligated to the ends of cDNA fragments. These were amplified for 12 cycles using primers incorporating unique index tags. Fragments were sequenced on an Illumina HiSeq 2500 or HiSeq 3000 using single reads extending 50 bases. Twenty-five to 30 million reads per library were targeted.

### 2.4. RNA-Seq Data Post-Processing and Normalisation

Raw reads from Fastq files from the sequencing runs underwent adapter trimming using the TrimGalore package (Barbraham Informatics, Cambridge, UK), version 0.6.2 [8]. Fastq files were then checked for quality using FastQC version 2 [9]. All files passed initial quality control. Reads were aligned using Salmon version 0.8.2 to the ENSEMBL reference human transcriptome, version 37.75, hg19 [10]. Reads were aligned using the --gcbias filter. Following alignment of reads, the txImport package, version 1.16.1, was used to import the reads into an R script from which further analyses were carried out [11]. Default parameter settings were used for all alignment and count generation steps.

Genes with low expression across samples were removed from further analysis. Specifically, genes with expression less than 1 TPM in fewer than 2 samples were removed from further analysis. Genes with coefficients of variation below the 25th percentile as compared to all genes were removed from further analysis. We then examined for batch effects using principal components analysis and observed a batch effect present in the data (Appendix A), that was corrected (Appendix A) by the Limma R package function remove Batch Effect, package version 3.44.3 [12].

Correlation analysis of gene expression data between primary and recurrent tumours was carried out using the Spearman correlation on batch-corrected, filtered, normalised mRNA expression.

### 2.5. Analysis of Tumour Subtypes

Pathway analysis was carried out using the GSVA package version 1.36.2 in R version 4.0, using single sample gene set enrichment analysis (ssGSEA) with gene sets as defined by Verhaak et al. [13,14]. The dominant molecular pathway (i.e., with the greatest ssGSEA score for the Verhaak mesenchymal, proneural, and classical signatures) was considered to be the samples’ molecular subtype.

### 2.6. Differential Expression Analysis

As noted above, transcript abundances were summarised to the gene level using the R package txImport, version 1.16.1. Gene-level count data were then filtered for low counts, such that only genes with a count of at least 1 in at least 2 samples were retained for further analyses. The resultant counts matrix was used as input to differential expression analysis with the DESeq2 R package, version 1.28.1 [15]. Covariates considered were the individual patient (if paired analysis was performed) and EGFR amplification status, where applicable. The threshold for statistical significance was set at Benjamini–Hochberg adjusted *p* < 0.05.

### 2.7. Pathway Expression among Primary and Recurrent Glioblastoma Samples

ssGSEA (single sample gene set enrichment analysis) as implemented by the GSVA package, version 1.36.2, in R version 4.0 was used to quantify activity of gene pathways. Pathways used were a subset of the MSigDB pathways as implemented in R by the msigdbr package, version 7.1.1 [16].

### 2.8. Survival Analyses

All survival analyses were performed in R using the Cox proportional hazards model with the survival package, version 3.1. Age and Karnofsky performance status (KPS) were considered to be continuous variables and were used as adjusting covariates in all multivariate survival analyses. Survival curves presented are based on the Kaplan–Meier estimator. Log rank testing was performed using the nph package, version 2.0.

### 2.9. Validation Cohort

In silico validation was performed using samples of matched primary and recurrent glioblastoma from the GLASS consortium dataset, data release 31 May 2022. Gene expression TPM and copy number data were obtained through the Synapse data release (syn17038081) and were analysed for the relationship between EGFR copy number and EGFR gene expression in both primary and recurrent samples.

## 3. Results

### 3.1. Cohort Characteristics

Glioblastoma samples from forty patients with matched primary and recurrent tumour specimens were characterised by RNA sequencing (total 80 samples from 40 individuals). 30–50 million reads were produced per sample, with a mapping rate of 25–40% across samples. Study design and patient clinical course is summarised in Figure 1a,b. Patients were followed for a median of 753.4 days (IQR 344.25–958.5). Complete clinical characteristics are outlined in Table 1. For each patient’s primary or recurrent tumour, EGFR amplification status was determined by fluorescence in-situ hybridisation (FISH). Tumour samples were characterised from the time of diagnosis (primary) and the time of first recurrence (recurrent). Median age at the time of diagnosis was 56.8 years (IQR 48.5–66.5), slightly less than the typical patient with glioblastoma (median age at diagnosis 65 years) [17], which is likely due to the need for re-resection and therefore slightly higher functional status. A majority of patients were males (28/40), and the median KPS was 90. The most common anti-tumour therapies administered to patients in this study were temozolomide (34/40), bevacizumab (13/40), lomustine (13/40), and tyrosine kinase inhibitors (3/40). Tyrosine kinase inhibitors used were dovitinib, lapatinib, and erlotinib in one patient each.

In our series, the presence of the EGFR amplification did not portend worse survival (*p* = 0.6, log-rank test; Figure 1c; *p* = 0.50 after adjustment for performance status and age at diagnosis). There was no statistically significant difference in the number of adjuvant temozolomide cycles received by patients with the EGFR amplification (3.05 cycles) and those without the EGFR amplification (3.21 cycles; *p* = 0.64, Wilcoxon rank sum test). Temozolomide given concurrently with radiotherapy was statistically significantly (*p* = 0.016) associated with better survival (HR 0.41, 95% CI 0.2–0.85) in an age-adjusted multivariate Cox proportional hazards model, but the number of adjuvant temozolomide cycles received was not statistically significantly predictive of overall survival (*p* = 0.22), even when EGFR amplification status was considered (*p* = 0.22). MGMT status was characterised in just 11/40 samples and was excluded from survival analysis due to underpowered analysis, as samples were collected in the era pre-MGMT testing and were not available for testing at the time of study. Notably, IDH1/2 mutation status was also not available for the same reason.

### 3.2. EGFR Amplification Remains Stable in Recurrence

Gene expression compared between primary and recurrent tumour samples was strongly correlated (Spearman’s rho = 0.79, *p* < 10^−15^, Appendix A). Among individual pairs of primary and recurrent tumours, mRNA expression was strongly positively correlated (Spearman’s rho = 0.64–0.91, *p* < 10^−15^ across all patients). The Spearman correlation coefficient between primary and recurrent gene expression did not associate with EGFR status (*p* = 0.49, two-sided Wilcox test), number of adjuvant TMZ cycles received (Spearman’s rho = −0.15, *p* = 0.36), or overall survival (Spearman’s rho = −0.25, *p* = 0.12). The correlation coefficient for gene expression in primary vs. recurrent tumours showed strong positive correlation with maximum tumour diameter (Spearman’s rho = 0.46, *p* = 0.0059), suggesting that differences in gene expression from primary to recurrence are more marked for larger tumours as compared to smaller tumours.

For the 39 of 40 samples in which initial EGFR amplification status was definitive (one primary sample was indeterminate), the EGFR amplification was statistically significantly more likely to be present in both primary and recurrent samples than lost or gained at the time of recurrence (*p* = 8.75 × 10^−5^, two-sided Fisher exact test; Figure 2a). In four of forty patients, EGFR was reported as amplified in the primary tumour, and non-amplified at recurrence. Among these four patients, all had received standard chemoradiotherapy with 0–7 cycles of adjuvant temozolomide after maximum surgical resection. None of these four patients received targeted anti-EGFR therapies. All patients who received targeted therapies to EGFR (*n* = 3, therapies were dovitinib, lapatinib, and erlotinib), retained the EGFR amplification in recurrence. Conversely, three patients did not have the EGFR amplification detected in the primary tumour but were found to have the EGFR amplification at the time of recurrence. In these three patients at the time of recurrence, the proportions of EGFR-amplified tumour cells were 18% (0 cycles of adjuvant temozolomide), 20% (6 cycles of adjuvant temozolomide), and 68% (0 cycles of adjuvant temozolomide), indicating that an amplified subpopulation may have been present at a lower clonal frequency initially.

The dominant molecular subtype as one of classical, proneural, or mesenchymal, was determined using the single sample gene set enrichment analysis (ssGSEA) score with mRNA gene signatures as originally defined by Verhaak et al. [13] Figure 2b depicts the distribution of these molecular subtypes between primary and recurrent samples. Most tumours (26/40) did not change molecular subtype, with the exception of proneural tumours, which were uncommon at the time of recurrence (5/40). Four of five tumours that were proneural at time of diagnosis were classical at recurrence. Six tumours that were of the classical subtype at the time of diagnosis were mesenchymal at the time of recurrence, and four tumours that were mesenchymal at the time of diagnosis were classical at recurrence. Of the fourteen tumours that switched molecular subtypes between the primary and recurrent settings, six were initially EGFR-amplified, and eight were initially EGFR non-amplified. EGFR-amplified tumours that switched subtype switched in three cases from classical to mesenchymal subtype, two cases from mesenchymal to classical subtype, and one case switched from proneural to classical subtype. Of the EGFR non-amplified tumours, three switched from proneural to classical, three switched from classical to mesenchymal, and two switched from mesenchymal to classical. EGFR-amplified tumours tended to be primarily classical in both primary (16/20) and recurrence (16/19), and EGFR non-amplified tumours tended to be classical (14/21) and mesenchymal (7/21) in recurrence.

### 3.3. Recurrent Tumours Show Elevated EGFR Pathway Activity

We compared EGFR-amplified and non-amplified tumours in the primary and recurrent settings independently. In differential expression analysis, of 20,922 total transcripts after filtering poorly characterised species, primary EGFR-amplified tumours had 180 transcripts statistically significantly downregulated as compared to non-amplified tumours, and 129 transcripts were upregulated (complete listing in Appendix A). In contrast, in the recurrent setting, of 19,580 total transcripts after filtering poorly characterised species, 60 transcripts were downregulated and 172 transcripts were upregulated between EGFR-amplified and non-amplified tumours (complete listing in Appendix A). Among these two groups of differentially expressed genes in primary and recurrent samples, 14 genes were common, and seven of these genes varied in a consistent direction with EGFR status (Figure 2c). Four genes increased concordantly in EGFR-amplified tumours in both primary and recurrent samples: EEA1 (co-localising early endosomal antigen 1 with the EGFR–EGF complex as it enters intracellularly [18]), IVD (isovaleryl-coA dehydrogenase), SEC61G (co-expressed with EGFR on chr 7p11 [19]), and EGFR itself. Three genes were concordantly decreased in EGFR-amplified versus non-amplified tumours: COL11A1, UBASH3B (ubiquitin associated and SH3 domain containing B, known to inhibit endocytosis of the EGF–EGFR complex [20]), and PPT2 (palmitoyl-protein thioesterase 2), visualised on volcano plots in Figure 3a,b.

As expected, the EGFR amplification associated with higher expression of EGFR mRNA in both primary and recurrent tumours (*p* = 0.00013 and *p* = 0.0017, respectively, Wilcoxon rank-sum test; Figure 2d). The Reactome EGFR pathway ssGSEA score, a measure of downstream EGFR pathway activity, was statistically significantly higher in primary tumours with the EGFR amplification (*p* = 0.033, Wilcoxon rank-sum test), but no statistically significant difference was seen in recurrent tumours with the EGFR amplification (*p* = 0.87, Wilcoxon rank-sum test; Figure 2e). In recurrent tumours, the EGFR pathway expression score for all patients, regardless of amplification status, was similar to that of the primary tumours with the EGFR amplification. This suggests that in recurrence, dysregulated mRNA expression may result in a state of functional EGFR pathway overexpression, suggesting its role in disease recurrence and maintenance.

We sought to validate our findings using the independent GLASS consortium dataset. We identified 121 primary and 147 recurrent tumours, 20 of which were EGFR-amplified in each group, with both copy number and RNA profiled. The pairing between primary and recurrent tumours in this dataset was not 1:1 as some tumours had multiple samples taken in recurrence, all of which were considered to be recurrent samples. Shown in Appendix A, the presence of the EGFR amplification was associated with greater mRNA expression levels of the EGFR transcript in both a primary and recurrent setting (*p* < 10^−3^, Wilcoxon rank sum test, in both cases), but was not associated with greater expression of EGFR pathway activity in either the primary (*p* = 0.41, Wilcoxon rank sum test) or recurrent (*p* = 0.78, Wilcoxon rank sum test) settings.

### 3.4. Recurrent EGFR-Amplified Tumours Display a Distinct Gene Expression Profile with Increased BRD2 Expression

In comparing paired primary and recurrent tumour samples, EGFR-amplified tumours displayed a unique set of differentially expressed genes when contrasted with EGFR non-amplified tumours. Among the EGFR-amplified tumours, of 18,609 transcripts after filtering poorly expressed species, 21 transcripts were differentially expressed (p_adj_ < 0.05, Appendix A). Six of these transcripts were overexpressed in recurrence, including the bromodomain gene BRD2 (p_adj_ = 9.3 × 10^−3^), thought to be essential to glioblastoma cell cycle maintenance [20] and an emerging therapeutic target [21,22]. Likewise, among the 15 downregulated transcripts in recurrence, ZIC2 (p_adj_ = 5.5 × 10^−4^), TFF1 (p_adj_ = 3.3 × 10^−9^), PRR3 (p_adj_ = 9.5 × 10^−7^), and LST1 (leukocyte-specific transcript 1, p_adj_ = 3.7 × 10^−3^) were noted (complete listing in Appendix A, volcano plot shown in Figure 3c). An entirely unique set of 114 transcripts (24 decreased in recurrence, 90 increased in recurrence) was found to be differentially expressed (p_adj_ < 0.05, Appendix A, volcano plot shown in Figure 3d) between primary and recurrent EGFR non-amplified tumours of the total 20,038 transcripts, suggesting that primary-recurrent tumour differences in transcriptome are generally distinct depending on EGFR amplification status.

Among the differentially expressed transcripts in EGFR non-amplified patients, the synaptic GABA receptor GABBR1 (p_adj_ = 3.0 × 10^−7^) transcript was increased in expression in recurrent tumours, as was the GABA receptor subunit GABRB1 (p_adj_ = 0.043) transcript; however, whether an increased number of transcripts results in more functional receptors will require further study. Consistent with this finding, it has been shown that GABA receptor functionality may decrease with increasing malignant potential of glial tumours [23]. Twenty-four transcripts were downregulated in recurrence among EGFR non-amplified tumours, and among these was TARDBP (p_adj_ = 0.034), coding for the neuronal protein TDP-43, involved in the transcriptional response in neurodegenerative disease via widespread RNA binding interactions [24].

## 4. Discussion

In this study, we present an analysis comparing the transcriptomes of primary and recurrent glioblastoma samples, stratified by EGFR amplification status. Our findings, taken together, reinforce that glioblastoma is incredibly heterogeneous, and that there are crucial biological differences between EGFR-amplified and non-amplified tumours. We also acknowledge that because study patients had tissue that was diagnostic of glioblastoma using the pre-WHO 2016 criteria, our cohort is representative of a mix of present-day glioblastoma and WHO grade 4 astrocytoma, which may limit the generalisability of our findings. IDH mutations were not routinely tested, and MGMT status is unknown for the majority of patients in this cohort. In addition, RNA-sequencing data were obtained from formalin-fixed paraffin-embedded specimens, impacting RNA quality prior to sequencing. Future studies would benefit from the use of fresh, frozen tissue prior to RNA sequencing, whenever possible.

We show that the landscape of recurrent glioblastoma is unique, wherein tumours display increased EGFR pathway expression without evidence of EGFR amplification, suggesting a phenotypic change not captured at the level of EGFR copy number or EGFR mRNA expression, arguing for an alternative mechanism for increased pathway activity. While the Verhaak molecular subtype often switched between primary and recurrent tumours (14 of 40 cases), the rate and patterns of switching were not associated with EGFR status. Our results compare well with literature describing how and why the EGFR amplification in glioblastoma confers a unique biology. In particular, recurrent EGFR-amplified tumours showed increased expression of ZIC2, which is involved in increased cell proliferation [25], and increased expression of TFF1, which is involved in tumour cell migration [26]. EGFR-amplified tumours tended to retain their EGFR-amplified status in recurrence, even despite targeted treatment in three cases.

Indeed, in vivo studies of gefitinib have shown that although there was an effect on dephosphorylating EGFR in amplified tumours, there was no demonstrable effect on EGFR pathway activity, highlighting mechanisms for resistance that are independent of direct EGFR pathway activation [6]. Conversely, whether the EGFR amplification is prognostic has been a subject of debate, and a recent meta-analysis suggests that there is insufficient evidence to conclude that the EGFR amplification is a prognostic biomarker [27]. We hypothesise that this may be the case, despite the unique biology carried by this amplification, because of mechanisms other than genetic amplification causing EGFR pathway activation, suggesting that EGFR mRNA or protein-level analysis may be needed for patient stratification.

In summary, our results highlight the difficulty in reducing pathway activity to singular genomic alterations and demonstrate that this is particularly true in the case of EGFR in primary and recurrent glioblastoma, outlining the need to better understand the factors mediating gene pathway expression in these patients [28,29].

## Figures and Tables

**Figure 1 cancers-15-00670-f001:**
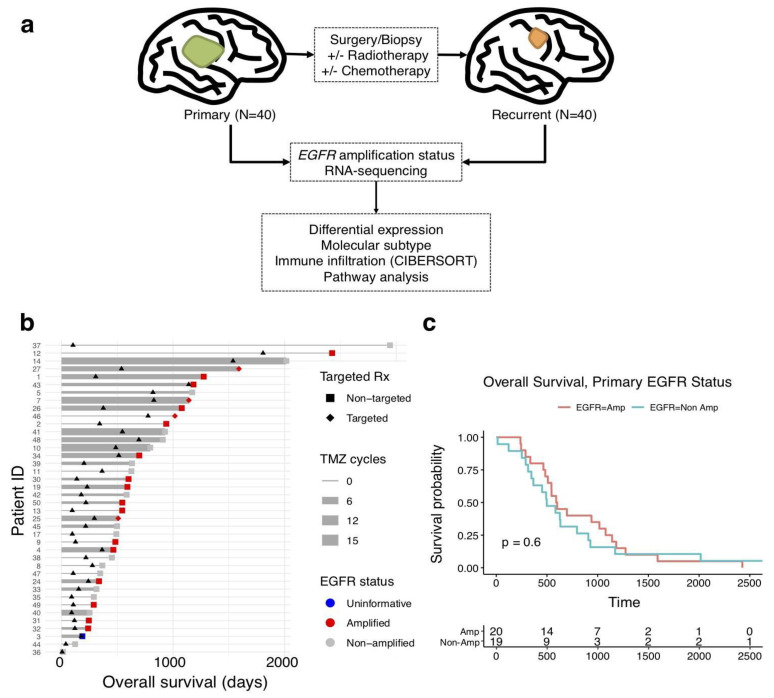
Study overview and participant characteristics. (**a**) Schematic illustration of study design. Samples were obtained from 40 patients with primary and recurrent glioblastoma; these were analysed by way of RNA sequencing and determination of the EGFR copy number. (**b**) The survival of the patients in the study cohort is depicted, with thicker bars indicating higher number of temozolomide (TMZ) cycles received and targeted therapy defined as small molecular inhibitors of EGFR. Triangles indicate the time of biopsy at recurrence. (**c**) Kaplan–Meier curves depicting survival between EGFR-amplified (red) and EGFR non-amplified (blue) cohorts.

**Figure 2 cancers-15-00670-f002:**
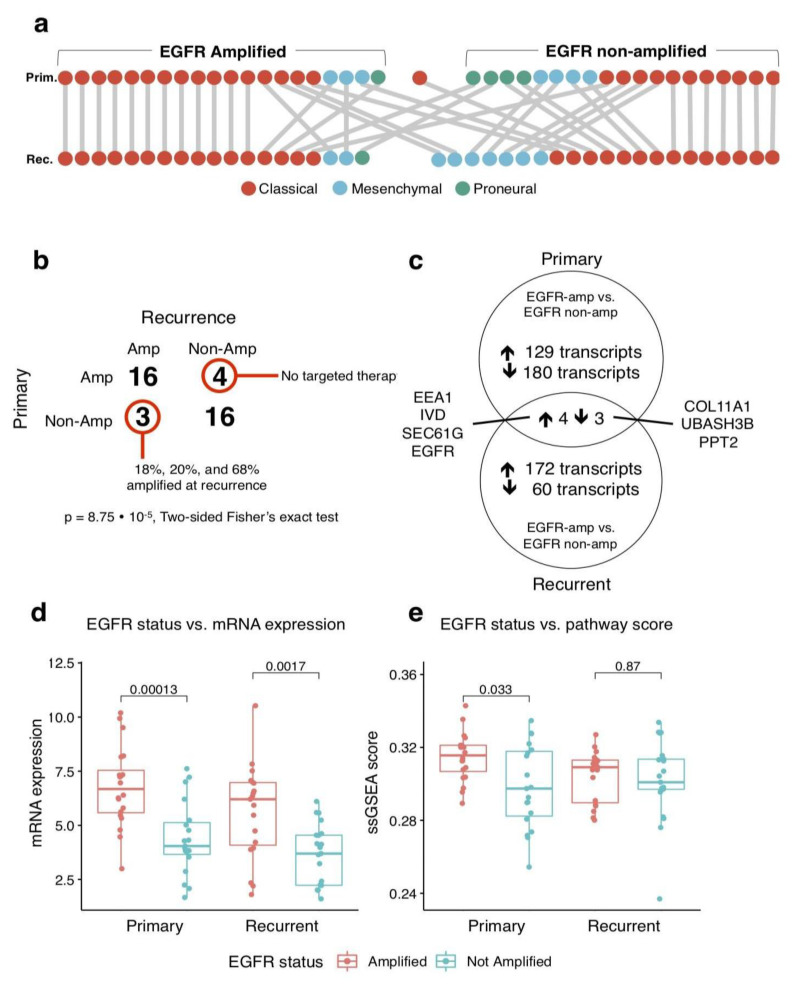
Overview of differences identified between EGFR-amplified and non-amplified tumours. (**a**) Schematic describing the changes in molecular subtype of each tumour between primary and recurrence, separated by EGFR amplification status. The singular primary tumour that is in neither group had indeterminate EGFR amplification status. (**b**) Two-by-two interaction table describing the relationship between EGFR amplification status in the primary tumour and recurrent tumour. (**c**) Differentially expressed transcripts between EGFR-amplified and non-amplified tumours in both primary (top circle) and recurrent tumours (bottom circle). Genes in common, either increased (left) or decreased (right) in EGFR-amplified tumours are listed in the centre of the diagram. (**d**) Box plots of EGFR mRNA expression (normalised, batch-corrected) with IQR boxed and median denoted by horizontal line, versus EGFR amplification status in the primary or recurrent tumour; *p* value is denoted by two-sided Wilcoxon rank sum test. (**e**) Analogous box plots describing EGFR gene signature score versus EGFR amplification status in primary and recurrent tumours.

**Figure 3 cancers-15-00670-f003:**
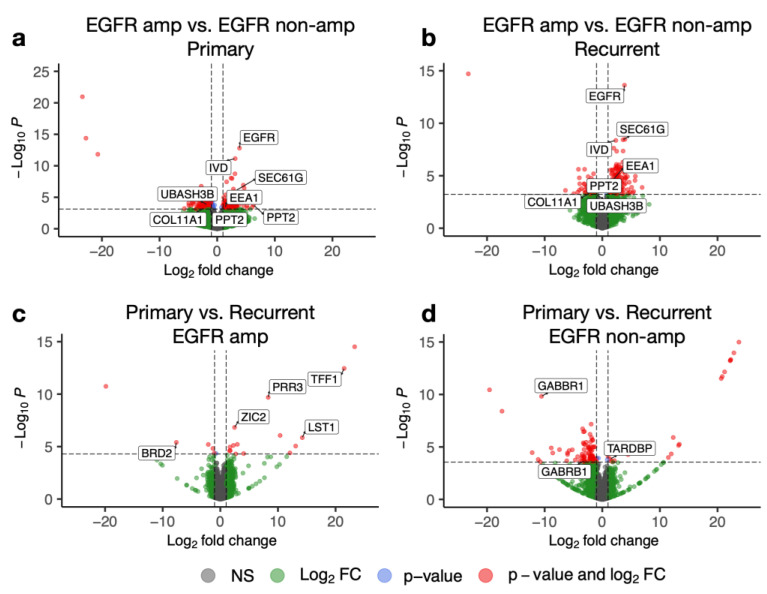
Volcano plots for differential gene expression for patient subgroups with key genes labelled. (**a**) Differential gene expression depicted for EGFR-amplified versus EGFR non-amplified primary tumours. (**b**) Differential gene expression depicted for EGFR-amplified versus EGFR non-amplified recurrent tumours. (**c**) Differential gene expression depicted for primary versus recurrent EGFR-amplified tumours. (**d**) Differential gene expression depicted for primary versus recurrent EGFR non-amplified tumours.

**Table 1 cancers-15-00670-t001:** Patient characteristics. Summary of patient characteristics for matched primary and recurrent tumour samples. IQR: interquartile range; KPS: Karnofsky performance status; NA: not available; PFS: progression free survival; OS: overall survival.

Primary Tumour EGFR Status	EGFR-Amplified *n* = 20	EGFR Non-Amplified/Indeterminate *n* = 20
Age at diagnosis (y)	56.9 (IQR 50–64.25)	56.7 (IQR48.5–67.25)
Sex	Male (13/20) Female (7/20)	Male (15/20) Female (5/20)
KPS at time of diagnosis	86.3 (IQR 80–90)	86.1 (IQR 80–90)
KPS at time of recurrence	76.3 (IQR 70–85)	79.4 (IQR 77.5–82.5)
Extent of primary resection	Total (10/20) Near total (7/20) Subtotal (2/20) Partial (1/20)	Total (13/20) Near total (4/20) Subtotal (2/20) NA (1/20)
*IDH1* mutation status	Wildtype (8/20) Mutant (1/20) NA (11/20)	Wildtype (5/20) Mutant (0/20) NA (15/20)
*MGMT* promoter methylation status	Unmethylated (6/20) Methylated (2/20) NA (12/20)	Unmethylated (2/20) Methylated (2/20) NA (16/20)
PFS (Median)	433.9 days (IQR 134.25–520 days)	320.2 days (IQR 101.5–394.75 days)
OS (Median)	809.75 days (IQR 478.5–1094 days)	697.1 days (IQR 308–824 days)
Laterality	Right (15/20) Left (5/20)	Right (13/20) Left (7/20)
Location	Temporal (7/20) Frontal (6/20) Parietal (6/20) Fronto-parietal (1/20)	Temporal (9/20) Frontal (6/20) Parietal (4/20) Parieto-occipital (1/20)
Targeted therapy received?	Yes (3/20)	

## Data Availability

Raw RNA seq data have been deposited into Gene Expression Omnibus (GEO) with accession GSE222515. All code will be available for download on GitHub at https://github.com/andrewdhawan/egfr-gbm-analysis.

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
