# Peer review of "EGFR Pathway Expression Persists in Recurrent Glioblastoma Independent of Amplification Status"

_cancers, 2023, doi:10.3390/cancers15030670_

Round 1

Reviewer 1 Report

EGFR amplification is commonly present in multiple cancer types including glioblastoma. Previous studies showed that EGFR-amplified tumors were more aggressive and resistant to the therapies. In this study, the authors characterized 80 glioblastoma samples with match primary and recurrent tumors. The authors measured the expression of EGFR and the gene pathway at the RNA level and associated these with the EGFR amplification status. The research topic was interesting and datasets generated from this study could be useful. However, the results were poorly presented and there are some issues that need to be addressed.

1.     In the method section, for the bioinformatics analysis, the authors should specify the parameters that were used. If the default parameter settings were used, the authors should clarify it.

2.     Section 3.1: “Median age at the time of diagnosis was 56.8 years (IQR 48.5-66.5), slightly less than the typical patient with glioblastoma”. What was the typical age of patients with glioblastoma? Was there any reference for this information?

3.     The authors should briefly describe the general RNA sequencing results, for example, how many reads were obtained in each sample, and what was the mapping rate. How many genes were selected after filtering steps for differential expression analysis?

4.     “Interestingly, this measure of genomic divergence showed strong positive correlation with maximum tumour diameter” How did the authors measure the genomic divergence?

5.     “In three of forty patients, EGFR was reported as amplified in the primary tumour, and non-amplified at recurrence.” In figure 2b, it showed 4 as amplified in the primary tumour, and non-amplified at recurrence.

6.     Section 3.3, the authors should present the RNAseq results in a better way. Figure 2c was oversimplified. The authors should consider using the MA plot and/or Volcano plot to show the differential expression in primary and recurrent data.

7.     In many places of this manuscript, the data in the main text was not consistent with the data in the figure. For example, “primary EGFR amplified tumours had 180 transcripts statistically significantly upregulated as compared to non-amplified tumours, and 129 transcripts were downregulated”. However, this the figure 2c, there were 129 with up arrow and 180 with down arrow. Same problem with the recurrent dataset.

8.     “As expected, the EGFR amplification associated with higher expression of EGFR mRNA in both primary and recurrent tumours (p = 0.00023…”  In figure 2d, p= 0.00013. Similarly, “was statistically significantly higher in primary tumours with the EGFR amplification (p = 0.02…” In figure 2e, p=0.033. Although these numbers won’t change the conclusion, the authors have the responsibility to make sure all the data and figures are accurately presented.

9.     The authors should consider presenting the results of section 3.4 in the main figure. It could be an interesting result for some readers. Also, this manuscript only had two main figures for the current version.

10.  The supplementary material for the section 3.4 was very confusing. In supplementary table 3 and 4, the increased genes had negative log2 fold change, and the decreased genes had positive log2 fold change. Also, the gene number listed in the tables were not consistent with the numbers that were described in the text.

11.  Code availability: please specify the github address.

12.  Data availability: please specify data accession numbers.

Reviewer 2 Report

To the authors:

Overall this paper shows increased EGFR pathway activity in recurrent glioblastoma and grade 4 astrocytomas regardless of EGFR amplification status. There are some minor errors within the paper and I have a few questions:

 Specific Comments:

Abstract:

Results: The sentence “Their molecular and clinical did not significantly differ from patients who received conventional chemotherapy” needs clarification, the molecular [what] and clinical [what] did not significantly differ? Profiles? Course? Outcomes?

Key Points: Grammatical issue - need a 1. before the first key point.

Introduction:

“Due to the lack of re-resection and re-biopsy in standard clinical practice, sufficient numbers of samples are not available to answer questions about the behaviour of glioblastoma during the course of therapy.”

This statement is not correct, the current guidelines for progressive glioblastoma recommend cytoreductive surgery in symptomatic patients and to improve overall survival.

Results –

Patient characteristics – IDH mutant status was unknown in 11/20 of the EGFR amplified and 15/20 of EGFR non-amplified. IDH mutant status is part of the diagnosis for Glioblastoma based on the WHO 2021 Classification, the tumors presented here are a mix of glioblastoma and grade 4 astrocytoma.

The header “3.2. EGFR amplification and GBM molecular subtype remains stable in recurrence” is not correct. In your sample of 40 patients, 11 changed molecular subtype based on Figure 2. Also the GBM molecular subtype is only mentioned in this one portion of the paper, is it relevant to your overall results and should be in the discussion?

Round 2

Reviewer 1 Report

The author addressed most of my previous questions.  In the revised manuscript, the authors claimed "30-50 million reads were produced per sample, with a mapping rate of 25-40% across samples." Good RNA-seq data should have ~90% mapping rate, why the data from this manuscript only had 25%-40% mapping rate? The entire manuscript was about the data analysis, if the data itself was of low quality, then the downstream analysis was problematic.

Round 3

Reviewer 1 Report

The authors answered my questions. The manuscript can be accepted after polishing the language.